# Better Understanding of Hydrogen Pellet Ablation Cloud Spectra through the Occupation Probability Formalism in LHD

**Motoshi Goto [1,\*](image), Gen Motojima [1], Ryuichi Sakamoto [1], Bernard Pégourié [2], Akinobu Matsuyama [3], Tetsutarou Oishi [4](image), Tomoko Kawate [1](image) and Yasuko Kawamoto [1]**

[1] National Institute for Fusion Science, Toki 509-5292, Japan; sakamoto@nifs.ac.jp (R.S.); kawate.tomoko@nifs.ac.jp (T.K.); kawamoto.yasuko@nifs.ac.jp (Y.K.)

[2] CEA, IRFM, F-13108 Saint-Paul-lez-Durance, France

[3] Graduate School of Energy Science, Kyoto University, Kyoto 611-0011, Japan; matsuyama.akinobu.3p@kyoto-u.ac.jp

[4] Department of Quantum Science and Energy Engineering, Tohoku University, Sendai 980-8579, Japan; tetsutarou.oishi.a4@tohoku.ac.jp

\* Correspondence: goto.motoshi@nifs.ac.jp

**Abstract:** We have recently incorporated the occupation probability formalism (OPF) in the simulation model [C. Stehlé and S. Jacquemot, *Astron. Astrophys.* **271**, 348 (1993)] to have a smooth transition from discrete lines to continuum spectrum in the wavelength range near the Balmer series limit. We have analyzed spectra measured for the hydrogen pellet ablation cloud in the Large Helical Device with the revised model, and have found that the electron density in the ablation cloud has a close correlation with the electron temperature of the background plasma. This type of correlation is first confirmed in the present analysis and should give a new insight in the simulation studies of pellet ablation for the magnetically confined fusion plasma.

**Keywords:** hydrogen pellet; ablation cloud; spectroscopy; Stark broadening; local thermodynamic equilibrium; occupation probability formalism





## 1. Introduction

A pellet is a small piece of solid material injected at a few hundred m/s into the magnetically confined fusion plasma. For the purpose of fueling the plasma, an ice made of hydrogen is used [1]. The impurity pellet injection is also utilized mainly for the study of impurity transport [2,3]. The pellet is vaporized in the plasma and forms the so-called ablation cloud of a centimeter size in the cross-field direction and of a few tenths of a cm in length along the field. The vaporization rate increases strongly with the plasma temperature, and the vaporization is completed typically in several hundred microseconds.

We have previously reported that the complete LTE (local thermodynamic equilibrium) is established in the ablation cloud of a hydrogen pellet injected in LHD (Large Helical Device) [4], and have demonstrated that representative parameters such as the electron temperature $T_e$ and the electron density $n_e$ in the ablation cloud can be determined through the fitting of measured UV-visible spectra with a simulation model which assumes the complete LTE condition.

This type of research has been extended to a spatial distribution measurement of $T_e$ and $n_e$ in the ablation cloud, where two-dimensional imaging measurements were conducted with three interference band-pass filters centered at different wavelengths and with different band widths [5]. Two of these filters used for $n_e$ determination were centered at the Balmer $\beta$ center, but had different band widths (half width) of 5 nm and 20 nm, respectively. The third one, used mainly for $T_e$ determination, was centered at 576.8 nm to measure the continuum radiation and had a band half width of 50 nm. We investigated the relationships among $T_e$, $n_e$, and the filtered intensity ratios by using a

complete LTE spectral model in advance so that $T_e$ and $n_e$ could be readily evaluated with the measurement results.

It has also been reported that the temporal developments of $T_e$ and $n_e$ in the ablation cloud show different characteristics when the magnetic field configuration for plasma confinement, i.e., the magnetic axis position and the magnetic field strength, is changed [6]. This magnetic field configuration dependence is thought to arise through $T_e$ and $n_e$ profiles of the background plasma and the local plasma confinement condition.

In these previous studies, when a fitting of the measured spectrum with a theoretical model was conducted, the transitional wavelength region from discrete lines to continuum near the Balmer-series limit, i.e., roughly 365 nm to 400 nm, was excluded because the spectrum in this region was difficult to construct and looked to be unimportant for determining the parameters. Actually, in many cases, the synthetic spectra constructed with the obtained fitting parameters agreed well with the measurements except for the wavelength region excluded from the fitting.

For some cases, however, especially when $n_e$ is lower than $1 \times 10^{23} \, \mathrm{m}^{-3}$, we found that the synthetic spectra showed significant disagreement with the measured spectra even with using the parameters obtained as a solution of the fitting. This problem will be discussed in more detail in Section 2. Although the reason of this disagreement was not clearly understood, it seemed that the transitional region which was excluded in the fitting was important for a comprehensive understanding of the spectra.

On the other hand, a connection between discrete lines and continuum radiation spectra has been a long-standing subject in the theoretical atomic physics field [7]. In this paper we report our attempt at analyzing the measured spectra with a scheme of the occupation probability formalism (OPF).

This kind of detailed spectroscopic analysis of the pellet ablation cloud is only performed at LHD, making this research highly unique. Improvement of the simulation model naturally leads to accuracy enhancement of derived plasma parameters in the pellet ablation cloud, which can be used for examining reliability of the pellet ablation simulation model. The results obtained in this study should therefore contribute to making clear the pellet ablation mechanism and hence enhancement of the particle fuelling efficiency for the magnetically confined fusion plasma.

## 2. Experiment

The experiment has been conducted in the Large Helical Device (LHD) which is a heliotron-type machine for magnetic confinement fusion experiments. The major and the averaged minor radii for the present experiment are 3.6 m and 0.6 m, respectively. The hydrogen pellets are injected with high pressure helium gas from an outboard side port as shown in Figure 1. The typical injection speed is 1 km/s and the intense radiation is observed during a period of roughly 400 µs, which indicates that the penetration depth into the plasma is roughly 0.4 m.

The emitted light is gathered using an optical fiber with a core diameter of 100 µm (STU100, MITSUBISHI CABLE INDUSTRIES, Tokyo, Japan). No spatial resolution measurements were conducted in this study, and the radiation emitted from the entire volume of the ablation cloud is observed. Consequently, the results obtained can be considered as indicative of a representative portion of the ablation cloud. The light is guided to a spectrometer that has a focal length of 0.5 m, and is equipped with a 100 grooves/mm grating (500is, Chromex, Albuquerque, NM, USA). A CCD (charge coupled device) is used for recording the spectra. The CCD has 1024 pixels (horizontal) × 255 pixels (vertical) with each pixel size of 26 µm × 26 µm (DU-420V, Andor, Belfast, UK). The horizontal direction of the CCD is used to record the spectrum. Only the top several lines of the CCD with a width corresponding to the optical fiber diameter, i.e., about 4 lines, are exposed, and the electric charges accumulated continuously shift by one line vertically every 16 µs after receiving a start trigger signal. The vertical direction of the CCD is thus used for recording the temporal development of the spectra. Although the spectra are recorded every 16 µs,

the actual time resolution is lower due to the optical fiber height being larger than the CCD pixel height.

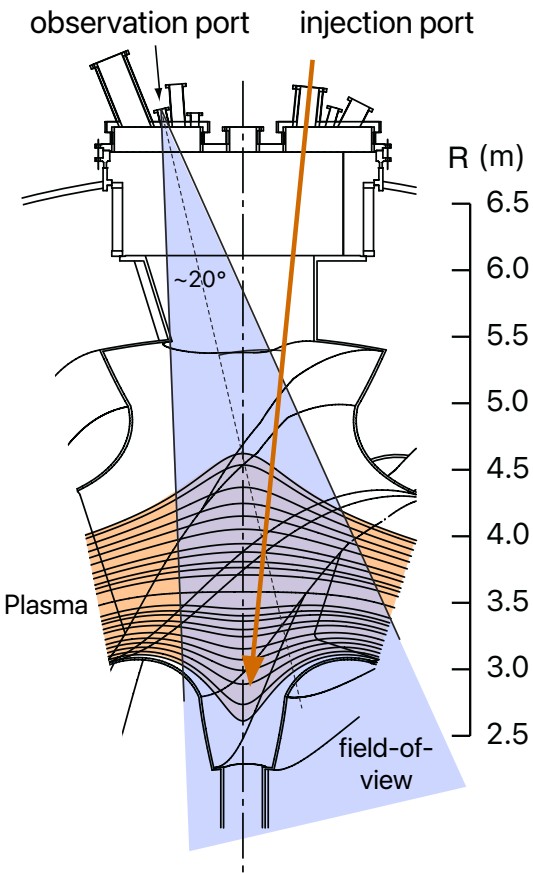

**Figure 1.** Top view of a LHD toroidal section for the pellet injection and the spectroscopic observations. The orange arrow denotes the trajectory of the pellet, while the orange-shaded region signifies the volume occupied by the plasma. The field of view is depicted by the light-blue area, and *R* shows the major radius.

Figure 2 shows examples of the measured spectra at different timings in the discharge of shot number 125838. The intensity is defined as the total radiation power from the ablation cloud per unit wavelength width. The spectra include some discrete Balmer series lines, which clearly show the Stark broadening. The broadening width of the discrete lines varies with time, which suggests that $n_e$ in the ablation cloud changes with time. The absence of a clearly visible dip at the center of the Balmer $\beta$ line can be attributed to the resolution of the spectrometer. In fact, the dip becomes distinctly noticeable when observed using another high-dispersion spectrometer. It is also noticed that the spectra include a significant amount of continuum radiation, which is known to be helpful to evaluate $T_e$ [4]. It is hence expected that the parameters $T_e$ and $n_e$ can be determined through a fitting of the measured spectra if we have an appropriate simulation model.

It is worth noting that the typical magnetic field strength at the location where the pellet primarily vaporizes is approximately 2 T, and the Zeeman splitting is estimated to be around 0.05 nm. This splitting is one order of magnitude smaller than the wavelength width corresponding to the detector's pixel size and remains unresolved with the current measurement system. Consequently, the Zeeman effect is not taken into account in the synthetic spectral model for this study.

Some recent studies have reported that the magnetic field could affect the line broadening [8–11]. However, the field strength in the present study is less than 3 T, which is so weak that such effects should be invisible and are not taken into account in the spectral model.

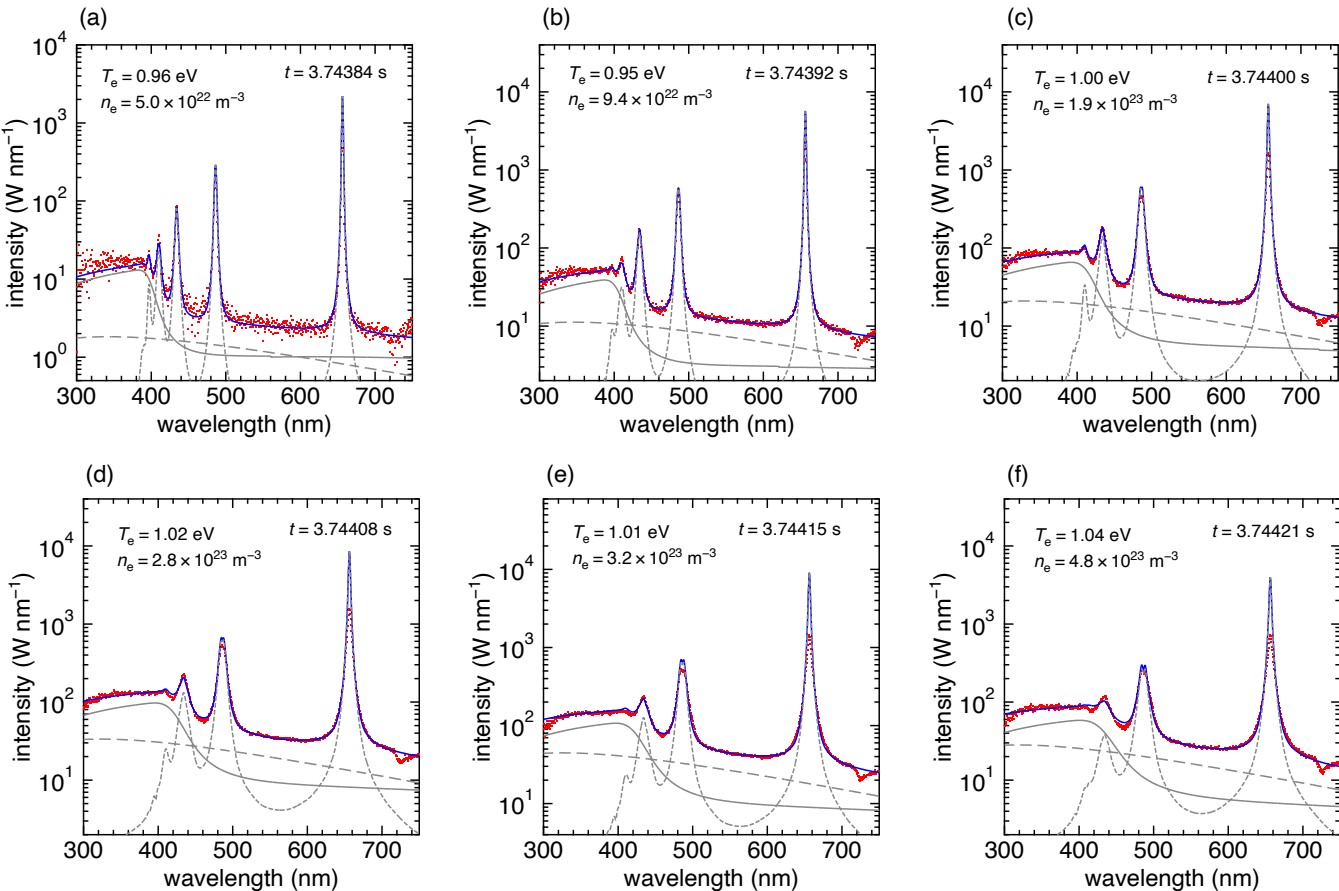

**Figure 2.** Examples of the spectra measured at different timings (**a**–**f**) of discharge 125838 (red dots). The actual measured times for (**a**–**f**) are given in the figures. The blue solid lines are the fitted results. The gray solid, long-dashed, and dashed lines show the radiative recombination continuum, the radiative attachment continuum, and the discrete line components, respectively.

Figure 3 shows an example of the fitting result (dotted curve) of a measured spectrum (red points) with a simulation model without the occupation probability formalism in the wavelength range near the Balmer series limit.

The wavelength range from 365 nm and 400 nm is excluded in the fitting. Although the fitted curve looks to agree well with the measured profiles for the Balmer $\gamma$ line at 434.046 nm and Balmer $\delta$ line at 410.173 nm, the Balmer $\varepsilon$ line at 397.007 nm and the continuum intensity in the range shorter than 365 nm show clear discrepancies with the measured spectrum. Therefore, the reliability of the derived parameters $T_e = 1.9$ eV and $n_e = 7.6 \times 10^{22}$ m$^{-3}$ are questionable, and modeling in this transitional region seems to be necessary for obtaining a better fitting result.

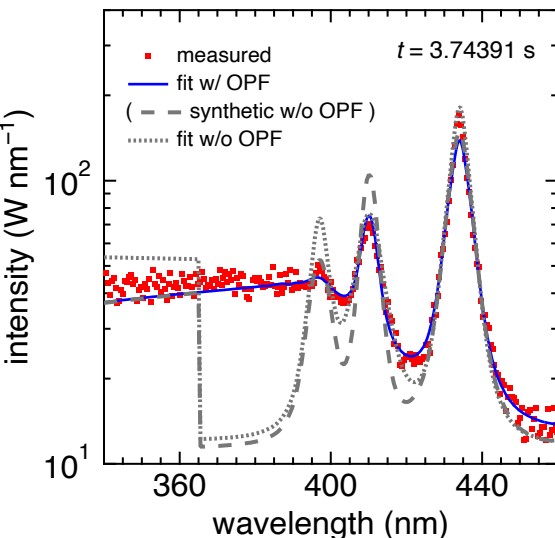

**Figure 3.** Example of fitting results of a measured spectrum (red dots) in the wavelength range near the Balmer series limit. The dotted and blue solid curves represent the results with a model without and with OPF, respectively. The  model without OPF gives $T_e = 1.9$ eV and $n_e = 7.6 \times 10^{22}$ m$^{-3}$, and the model with OPF gives $T_e = 0.95$ eV and $n_e = 1.0 \times 10^{23}$ m$^{-3}$. The dashed curve shows a synthetic spectrum with the model without OPF and the parameters obtained by the model with OPF.

## 3. Synthetic Spectral Model

For fitting the measured spectra, we have developed a synthetic spectral model where we assume a condition of complete LTE which was confirmed in the previous study [4]. A spectrum is regarded as a superposition of discrete lines and continuum radiation, and under the complete LTE, the entire spectrum except the absolute magnitude can be calculated only with $T_e$ and $n_e$. It is noted that the magnitude corresponds to the volume of the ablation cloud. In the present study, we have revised the model by incorporating the occupation probability formalism (OPF), which works to smoothly connect the spectra dominated by the discrete lines and by the continuum radiation in the wavelength region including the Balmer series limit.

### 3.1. Discrete Lines

The discrete line intensities are proportional to the corresponding upper level populations of the transitions. The level population $n(p)$ conforms to the Saha–Boltzmann equation [12]

$$n(p) = p^2 \left( \frac{h^2}{2\pi mkT_e} \right)^{3/2} \exp\left( \frac{R}{p^2 kT_e} \right) n_e n_i, \tag{1}$$

where $p$ represents the principal quantum number, $h$, $m$, $k$, and $R$ are the Planck constant, the electron mass, the Boltzmann constant, and the Rydberg constant, respectively, and $n_i$ is the proton density. Each line profile is assumed to be dominated by Stark broadening in the analysis, for which we use theoretical data published as a part of ref. [13]. This dataset provides numerical line profile data at some discrete $T_e$ and $n_e$ sets. Each line profile data consists of normalized intensities at some wavelengths relative to the line center. We obtain line profiles by resampling for the wavelength and by interpolation for $T_e$ and $n_e$.

### 3.2. Continuum Radiation

The continuum radiation is known to be dominated by the radiative recombination continuum and the radiative attachment continuum [4]. The radiation power density can be expressed as

$$\rho_p^R(\nu)d\nu = h\nu n_i n_e f(\varepsilon) v_e \sigma_{rec}(p,\varepsilon) d\varepsilon \quad \text{Wm}^{-3}, \tag{2}$$

where $\sigma_{\rm rec}(p, \varepsilon)$ is the cross section of the radiative recombination for a free electron having the kinetic energy of $\varepsilon$ to the level $p$, $f(\varepsilon)$ is the energy distribution function of free electrons, $v_{\rm e} = \sqrt{2E/m}$, and $\mathrm{d}\varepsilon = h\mathrm{d}v$.

The cross section $\sigma_{\rm rec}(p, \varepsilon)$ is derived from that of the photoionization $\sigma_{\rm ion}(p, v)$ and Milne's formula [12] as

$$\frac{\sigma_{\rm ion}(p, v)}{\sigma_{\rm rec}(p, \varepsilon)} = \frac{2Md^2\varepsilon}{h^2v^2}\frac{g_{\rm H^+}}{g_{\rm H}(p)}, \tag{3}$$

where $c$ is light speed and $g_{\rm H^+}(= 1)$ and $g_{\rm H}(p)(= 2p^2)$ are the statistical weights of the proton and the level $p$ of hydrogen atoms, respectively.

The Maxwell–Boltzmann distribution is here assumed for the electron velocity distribution function and the corresponding energy distribution

$$f(\varepsilon)\mathrm{d}\varepsilon = 2\sqrt{\frac{\varepsilon}{\pi}}\left(\frac{1}{kT_{\rm e}}\right)^{3/2}\exp\left(-\frac{\varepsilon}{kT_{\rm e}}\right)\mathrm{d}\varepsilon \tag{4}$$

is used.

Equation (2) can be then rewritten as

$$\rho_p^{\rm R}(v) = 2\sqrt{\frac{2}{\pi}}\frac{h^4}{m^{3/2}c^2}p^2\left(\frac{1}{kT_{\rm e}}\right)^{3/2}\exp\left(-\frac{hv - \chi_{\rm H}(p)}{kT_{\rm e}}\right)\sigma_{\rm ion}(p, v)v^3 n_{\rm i} n_{\rm e} \quad \text{Wm}^{-3}\text{s}, \tag{5}$$

where $\chi_{\rm H}(p)$ is the ionization potential of the level $p$.

The total radiation power due to the radiative recombination is obtained by taking the sum over the terminating level $p$ of the process. The expression is finally translated into a function of the wavelength $\lambda$ as

$$P^{\rm R}(\lambda) = \sum_p \rho_p^{\rm R}(v)\frac{v^2}{c} \quad \text{Wm}^{-4}. \tag{6}$$

where $\lambda = c/v$.

The radiative attachment is such a process that a free electron attaches to a neutral atom and a negative ion is created, which is expressed as

$$\mathrm{H} + \mathrm{e} \rightarrow \mathrm{H}^- + hv. \tag{7}$$

Because the attached electrons have a continuous energy spectrum, the continuum radiation is emitted by this process.

The radiation power density is expressed as

$$\rho^{\rm A}(v)\mathrm{d}v = hvn_{\rm H}n_{\rm e}f(\varepsilon)v\sigma_{\rm att}(\varepsilon)\mathrm{d}\varepsilon \quad \text{Wm}^{-3}, \tag{8}$$

where $n_{\rm H}$ is the hydrogen atom density. The cross section of the radiative attachment process $\sigma_{\rm att}(\varepsilon)$ can be obtained from the photodetachment cross section $\sigma_{\rm det}(\varepsilon)$ again with Milne's formula as

$$\frac{\sigma_{\rm det}(v)}{\sigma_{\rm att}(\varepsilon)} = \frac{2mc^2\varepsilon}{h^2v^2}\frac{g_{\rm H}(1)}{g_{\rm H^-}}, \tag{9}$$

where $g_{\rm H^-}(= 1)$ is the statistical weight of the ground state of the negative ion.

Equation (8) is then rewritten as

$$\rho^{\rm A}(v) = \sqrt{\frac{2}{\pi}}\frac{h^4}{m^{3/2}c^2}\frac{1}{2}\left(\frac{1}{kT_{\rm e}}\right)^{3/2}\exp\left(-\frac{hv - \chi_{\rm H^-}}{kT_{\rm e}}\right)\sigma_{\rm det}(v)v^3 n_{\rm H} n_{\rm e} \quad \text{Wm}^{-3}\text{s}, \tag{10}$$

and the total radiation power density spectrum due to the radiative attachment is expressed as a function of the wavelength as

$$P^{\mathrm{A}}(\lambda) = \rho^{\mathrm{A}}(\nu)\frac{\nu^2}{c} \quad \mathrm{Wm}^{-4}. \tag{11}$$

We adopt the data in ref. [14] for $\sigma_{\mathrm{det}}(\nu)$. It is noted that in our previous paper [4], we have verified that the negative ion density $n_{\mathrm{H}}^{+}$ is two orders of magnitude lower than $n_{\mathrm{e}}$ under the plasma conditions here considered and does not exert a significant influence on the results.

On the other hand, the contribution of the Bremsstrahlung is confirmed to be negligibly small as compared to other continuum radiation components under the present plasma condition.

*3.3. Occupation Probability Formalism (OPF)*

In principle, a combination of these three components, i.e., the discrete line profiles, the radiative recombination continuum, and the radiative attachment continuum, gives a complete spectrum which can be compared with the measurement result. However, it is immediately recognized that there is a discontinuity in the spectrum which is never detected in the actual observation. The discontinuity is created because the radiative recombination continuum is terminated at the wavelength corresponding to the ionization limit of the atomic hydrogen.

In our previous studies we have excluded the wavelength range near the recombination continuum limit when we fit the measured spectra with the synthetic model. However, we have found that in some cases the wavelength range where the continuum gradually changes to discrete lines is important for fitting the measured spectrum.

For solving this problem, we adopt OPF. We first think about an extreme condition where a discrete line is replaced with a flat continuum radiation spectrum having the same integrated intensity as the discrete line, i.e.,

$$\int_{\mathrm{line}} P_p^{\mathrm{L}}(\lambda)\mathrm{d}\lambda = \overline{P}_p^{\mathrm{L}}\delta\lambda, \tag{12}$$

where $\overline{P}_p^{\mathrm{L}}$ is the constant continuum intensity and $\delta\lambda$ is the wavelength width which belongs to a discrete line from level denoted by the principal quantum number $p$. The width $\delta\lambda$ can be evaluated as

$$\delta\lambda = \left|\frac{\mathrm{d}\lambda}{\mathrm{d}p}\right| = \frac{2\lambda}{\left(\frac{p^2}{4} - 1\right)p}, \tag{13}$$

where

$$\lambda = \frac{hc}{R\left(\frac{1}{2^2} - \frac{1}{p^2}\right)} \tag{14}$$

is used.

It is noted that such a virtual continuum radiation spectrum can be reasonably expressed with the same equation as the radiative recombination continuum with replacing the exponential factor $\exp[-\varepsilon/(kT_{\mathrm{e}})]$ with $\exp[\chi(p)/(kT_{\mathrm{e}})]$, where $\chi(p)$ is the ionization potential of level $p$ [12].

Under the OPF framework, the spectrum consists of two parts, i.e., the discrete line profile part and continuum radiation part like

$$I_p(\lambda) = w_{\mathrm{f}}I_p^{\mathrm{L}}(\lambda) + (1 - w_{\mathrm{f}})\overline{I}_p^{\mathrm{L}}, \tag{15}$$

where $w_{\mathrm{f}}$ is the fraction of the discrete part in the total intensity, which will be discussed later on.

The discrete part $I_p^{\mathrm{L}}(\lambda)$ is regarded as

$$I_p^{\mathrm{L}}(\lambda) = n(p)P_p(\lambda), \tag{16}$$

where $n(p)$ is the level $p$ population and $P_p(\lambda)$ is the line profile normalized as

$$\int P_p(\lambda)d\lambda = 1. \tag{17}$$

In the present case, $P_p(\lambda)$ is dominated by the Stark broadening as already mentioned.

The coefficient $w_f$ is derived with the microfield distribution in the ablation cloud and the critical electric field defined as the field ionization threshold of the state. Figure 4a shows an example of the Holtsmark distribution $P(F)$ for the cases of $n_e = 10^{21}\,\mathrm{m}^{-3}$, $10^{22}\,\mathrm{m}^{-3}$, and $10^{23}\,\mathrm{m}^{-3}$, where $P(F)$ is normalized as

$$\int_0^\infty P(F)dF = 1. \tag{18}$$

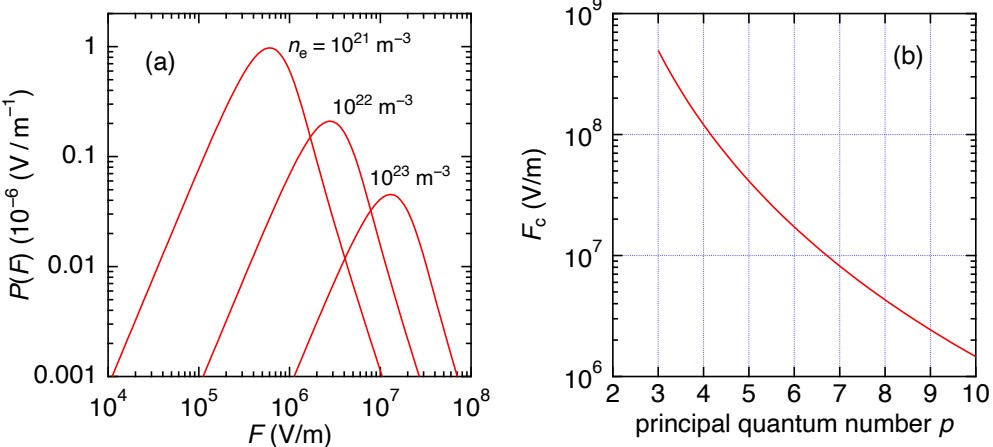

**Figure 4.** (**a**) Holtsmark distributions for $n_e = 10^{21}\,\mathrm{m}^{-3}$, $10^{22}\,\mathrm{m}^{-3}$, and $10^{23}\,\mathrm{m}^{-3}$. (**b**) Dependence of the critical field $F_c$ evaluated by Equation (19) on the principal quantum number $p$ of hydrogen atoms.

It should be here noted that under the present plasma condition, the ratio of the nearest distance of perturbers and the Debye radius is evaluated to be 0.57. In that case, the ion correlation effects become significant and, strictly speaking, the Holtsmark distribution should be modified. However, its impact on our results is limited and we adopt the Holtsmark distribution throughout.

The critical field $F_c$ is evaluated following ref. [15] as

$$F_c = \frac{2p - 3.5}{6p^3(p+1)^2(p-2)}, \qquad (\text{a. u.}) \tag{19}$$

and is plotted in Figure 4b.

We regard the integral of $P(F)$ in the range of $F < F_c$ corresponds to the fraction of discrete state, i.e.,

$$w_f = \int_0^{F_c} P(F)dF. \tag{20}$$

Figure 5 shows $w_f$ as a function of $p_{\mathrm{eff}}$ for several $n_e$ values, where $p_{\mathrm{eff}}$ represents the principal quantum number treated as a continuous variable.

In the case of $n_e = 10^{23}\,\mathrm{m}^{-3}$ and $p_{\mathrm{eff}} = 6$, $w_f$ is evaluated to be approximately 0.5, which means that a half of the radiation is treated as discrete lines and the remaining half as the continuum.

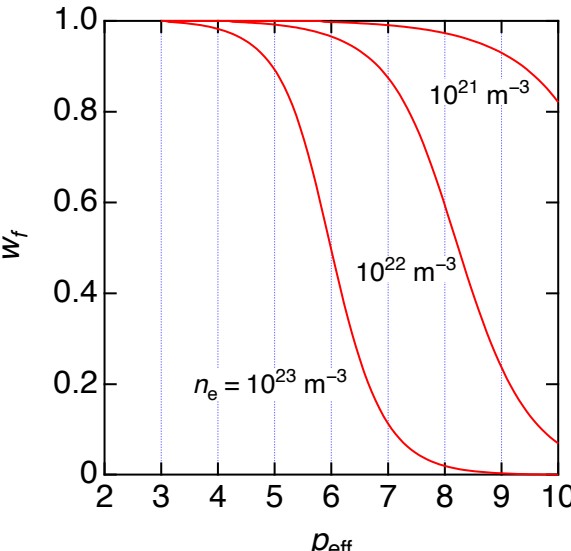

**Figure 5.** Dependence of the occupation probability $w_f$ on $p_{\mathrm{eff}}$ for $n_{\mathrm{e}} = 10^{21}\,\mathrm{m}^{-3}$, $10^{22}\,\mathrm{m}^{-3}$, and $10^{23}\,\mathrm{m}^{-3}$.

## 4. Results and Discussion

We have conducted least squares fittings for the measured spectra in Figure 2 with the synthetic spectral model discussed in the previous section. The wavelength ranges near the Balmer $\alpha$ and $\beta$ line centers are excluded in the fitting because these lines could be influenced by the reabsorption effect, which is not included in the model. We will attempt a quantitative analysis with a simple radiation transport model later on.

An example of the fitting is shown in Figure 3. The blue solid curve, which is the fitted spectrum with OPF, is in good agreement with the measured spectrum even in the transitional region. The derived parameters, i.e., $T_{\mathrm{e}} = 0.95\,\mathrm{eV}$ and $n_{\mathrm{e}} = 1.0 \times 10^{23}\,\mathrm{m}^{-3}$, are found to be significantly different to those obtained in the model without OPF. In the same figure, a synthetic spectrum by the model without OPF but with the parameters obtained with the OPF model is shown with the gray dashed line, which shows a significant discrepancy from the measurement in the wavelength range including the Balmer $\delta$ line. This result indicates that the OPF model is necessary to obtain reliable fitting results for the measured spectra.

The results of the fittings for all the spectra in Figure 2 are shown with the thick solid lines in Figure 2. Satisfactory agreements are obtained between the measured and synthetic spectra for all the cases. The spectra corresponding to Equations (6) and (11) are shown with the thin solid lines and the dashed lines, respectively. The radiative attachment continuum is shown by the dash-dotted line. The fitting parameters $T_{\mathrm{e}}$ and $n_{\mathrm{e}}$ derived are given in the figures.

It should be noted here that this paper focuses on the method to smoothly connect discrete lines to the continuum in the Balmer series spectrum, and the data for the Stark broadening profiles remain the same in the present analysis. However, around the same time as ref. [4], a new calculation method for the Stark broadening of the Balmer series lines has been developed [16,17]. This new method, called the frequency fluctuation method (FFM), has improved the treatment for the electron collision broadening by taking into account the gradual change from the impact approximation to the quasi-static approximation with increasing $n_{\mathrm{e}}$, which typically leads to narrowing the line profile. Following the criterion for this transition discussed in refs. [16,18], it is found that this effect for the Balmer $\alpha$ and $\beta$ lines is found to be insignificant with the present $n_{\mathrm{e}}$ and $T_{\mathrm{e}}$ ranges, while the influence on the Balmer $\gamma$ could be significant. Figure 2f is the spectrum when $n_{\mathrm{e}}$ is relatively high, and the measured Balmer $\gamma$ profile is found to be narrower than the

synthetic spectrum. This difference could be explained by FFM, but further investigation would be necessary.

Similar analyses have been made for the spectra taken at all the timings, and the obtained $T_e$ and $n_e$ are plotted against the major radius $R$ which is estimated from the time of the discharge indicated on the top axis in the lower panel of Figure 6.

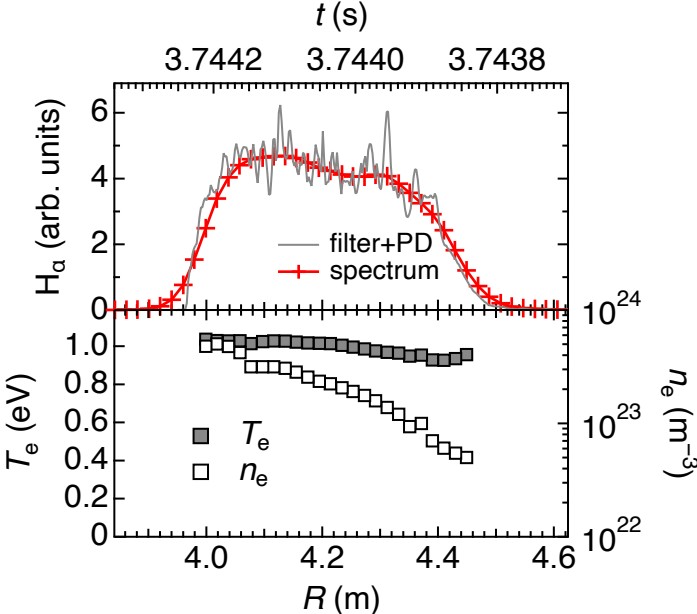

**Figure 6.** Radial profiles of the Balmer $\alpha$ line intensities measured by the filter + photodiode system (solid line) and by the spectral observation (plus symbols) (**upper panel**), and the parameters $T_e$ and $n_e$ obtained by the fitting (**lower panel**). The corresponding time of the discharge is indicated by the top axis.

It is found that the variation of $T_e$ is small while $n_e$ increases with time. It is noted that the start timing of the data acquisition has some uncertainty while the sampling time is known to be 16 μs, as mentioned in Section 2. Therefore, the absolute record time of the spectral data has been calibrated with the help of the other signal taken simultaneously. The gray line in the upper panel shows the Balmer $\alpha$ line intensity measured by a photodiode with an interference filter with the sampling rate of 500 kHz. The observation is made with an optical fiber which is located at the same port as the spectral measurement. The entire ablation process takes place in the field-of-view of this measurement. The signal is recorded under the common clock of the LHD experiment.

We have derived the temporal variation of the Balmer $\alpha$ line intensity from the spectral measurement and the absolute time is calibrated so that the center-of-mass of the intensity coincides with that of the photodiode signal. The result is shown in the upper panel of Figure 6 where signal intensity is normalized so that the intensity integrated over time is equal to that of the photodiode signal. Although the fine structures observed in the photodiode signal are missing in the spectral measurement due to the lower time resolution, the agreement of the global shapes between the two signals are satisfactory. The top axis shows the radial position $R$ of the ablation cloud evaluated under an assumption that the pellet is in linear motion at constant velocity where the velocity is measured by the time-of-flight technique in the guide tube. It is noted that because the pellet is injected from the outboard side, the radius of the pellet position decreases with time. The same data except the photodiode signal are replotted against $R$ with green symbols in Figure 7a–c.

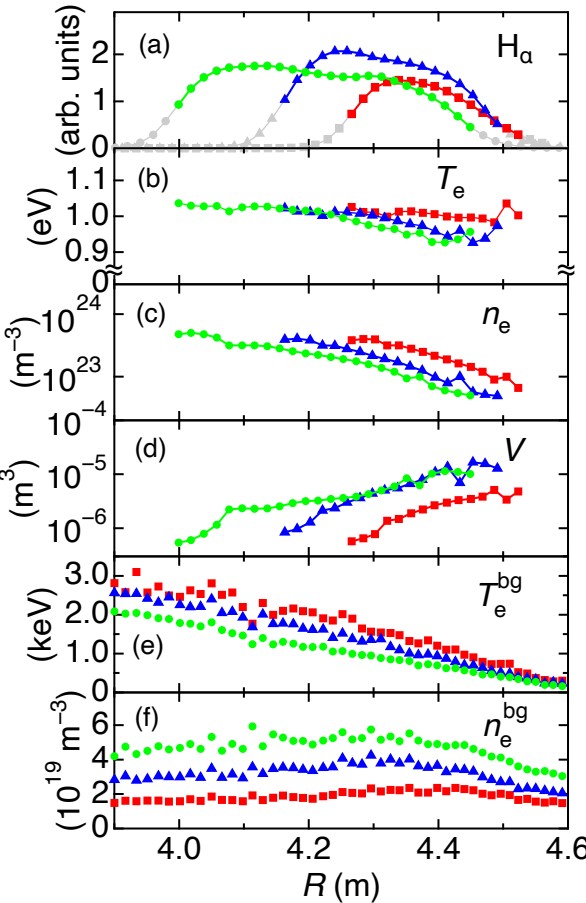

**Figure 7.** Radial profiles of (**a**) the Balmer $\alpha$ line intensity, (**b**) $T_e$, (**c**) $n_e$, (**d**) plasma volume $V$, (**e**) $T_e^{bg}$, and (**f**) $n_e^{bg}$ for three discharges, i.e., 125794 (squares), 125816 (triangles), and 125838 (circles). The gray areas of the Balmer $\alpha$ signals indicate that it is possible to evaluate the Balmer $\alpha$ intensity, but fitting the spectrum is difficult because the signals are too weak.

The same analysis has been made for two different discharges with the same magnetic configuration. The results are plotted with the red squares and blue triangles in Figure 7a–c and it is noticed that some differences are seen among the three cases. In the blue and red cases, the Balmer $\alpha$ intensity data suggests a shorter penetration depth than the green case. It is noted that the pellet is moving from right to left in this figure so that the penetration depth is measured from the right edge.

It is also noticed that $T_e$ changes little with time, and there is no major difference among the three discharges, except that the radial region being measured differs due to the difference in the penetration length. As for $n_e$, all three cases show a monotonically increasing behavior with the penetration depth in the entire region, and they finally arrive at similar maximum values even though the delay time to get the maximum is different. Figure 7d shows the plasma volume obtained as the scale factor in the fitting of the spectrum and they show a decreasing behavior. This is understandable because the Balmer $\alpha$ intensity is little changed while $n_e$ increases during the ablation. These results fit well with the simulation model, which shows that $T_e$ remains always close to 1 eV and $n_e$ increases sharply with $T_e$ through the vaporization rate.

Figure 7d,e show the electron temperature profiles $T_e^{bg}$ and the electron density profiles $n_e^{bg}$ of the background plasma, respectively, measured by the Thomson scattering diagnostic system. We continuously measure the $T_e^{bg}$ and $n_e^{bg}$ profiles at 33 ms intervals, and the data displayed in Figure 7d,e represent the values from the final measurement taken just before the pellet injection. These parameters in the background plasma are thought to be important for characterizing the ablation cloud condition. It is readily noticed that higher $T_e^{bg}$ leads to

shorter penetration depth, and the pellet cannot penetrate into the region with $T_{\mathrm{e}}^{\mathrm{bg}} > 2\,\mathrm{keV}$. It is also suggested that $n_{\mathrm{e}}$ has a close correlation with $T_{\mathrm{e}}^{\mathrm{bg}}$.

In Figure 8, $n_{\mathrm{e}}$ values are plotted against $T_{\mathrm{e}}^{\mathrm{bg}}$ and their close correlation is actually confirmed. In such a correlation between $n_{\mathrm{e}}$ and $T_{\mathrm{e}}^{\mathrm{bg}}$, fast ions originating from neutral beams could play a role because the fast ions are known to accelerate the vaporization process and the fast ion density is expected to be higher in lower $n_{\mathrm{e}}^{\mathrm{bg}}$ and hence higher $n_{\mathrm{e}}^{\mathrm{bg}}$ regions.

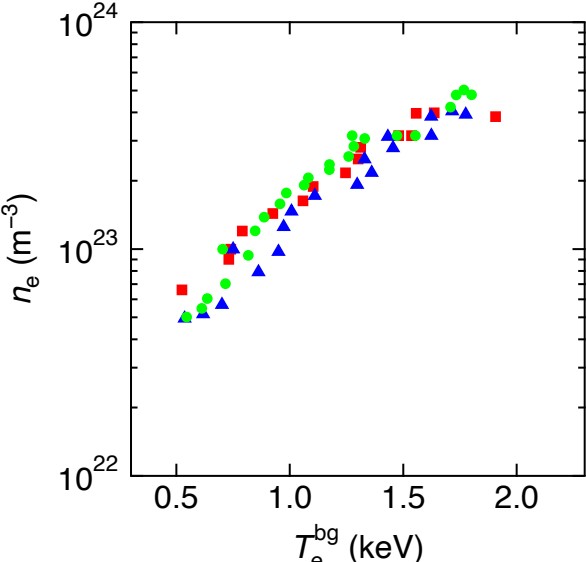

**Figure 8.** Dependence of $n_{\mathrm{e}}$ on $T_{\mathrm{e}}^{\mathrm{bg}}$ at the ablation cloud location for the three discharges in Figure 7. The same symbols as Figure 7 are used.

We here attempt an evaluation of the intensity reduction due to the reabsorption effect for the Balmer $\alpha$ line. The one-dimensional radiation transport is written as

$$\frac{\mathrm{d}I(\lambda, x)}{\mathrm{d}x} = -\kappa(\lambda)I(\lambda, x) + \eta(\lambda), \tag{21}$$

where $I(\lambda, x)$ is the intensity profile after passing through an absorption medium having a thickness of $x$, and $\eta(\lambda)$ and $\kappa(\lambda)$ are the emission coefficient and the absorption coefficient, respectively, which are assumed to be fixed in the medium.

The solution of Equation (21) can be expressed as

$$I(\lambda, x) = \frac{\eta(\lambda)}{\kappa(\lambda)}\{1 - \exp[-\kappa(\lambda)x]\}. \tag{22}$$

The coefficients $\eta(\lambda)$ and $\kappa(\lambda)$ are explicitly written as

$$\eta(\lambda) = \frac{hc}{4\pi\lambda}n(3)A(3,2)P(\lambda) \qquad \mathrm{Wm^{-3}sr^{-1}m^{-1}} \tag{23}$$

and

$$\kappa(\lambda) = \frac{hc}{4\pi\lambda}[n(2)B(2,3) - n(3)B(3,2)]P'(\lambda) \qquad \mathrm{m^{-1}}, \tag{24}$$

respectively, where $A(3,2)$, $B(2,3)$, and $B(3,2)$ are Einstein's coefficients in terms of the spectral intensity per unit wavelength interval [19] between $p = 2$ and $3$, and $P(\lambda)$ and $P'(\lambda)$ are the emission and absorption profiles normalized as $\int P(\lambda)\mathrm{d}\lambda = \int P'(\lambda)\mathrm{d}\lambda = 1$.

We here assume an identical profile for the emission and the absorption, i.e., $P(\lambda) = P'(\lambda)$. Under LTE, $\kappa(\lambda)$ can be rewritten as [4]

$$\kappa(\lambda) = \frac{\lambda^4}{8\pi c}\left[\exp\left(\frac{hc/\lambda}{kT_\mathrm{e}}\right) - 1\right]n(3)A(3,2)P(\lambda). \tag{25}$$

We fit the Balmer $\alpha$ spectrum in Figure 2c with Equation (22) by adjusting $x$. The best fit has been obtained with $x = 4.7 \times 10^{-3}$ m and the optical thickness, i.e., $\kappa(\lambda)x$ in Equation (22), at the line center is evaluated to be 6.54 in that condition. The resulting profile is shown with the blue solid line in Figure 9a, where the measured profile and the synthetic profile without the reabsorption effect are shown with the red dots and the dashed line, respectively.

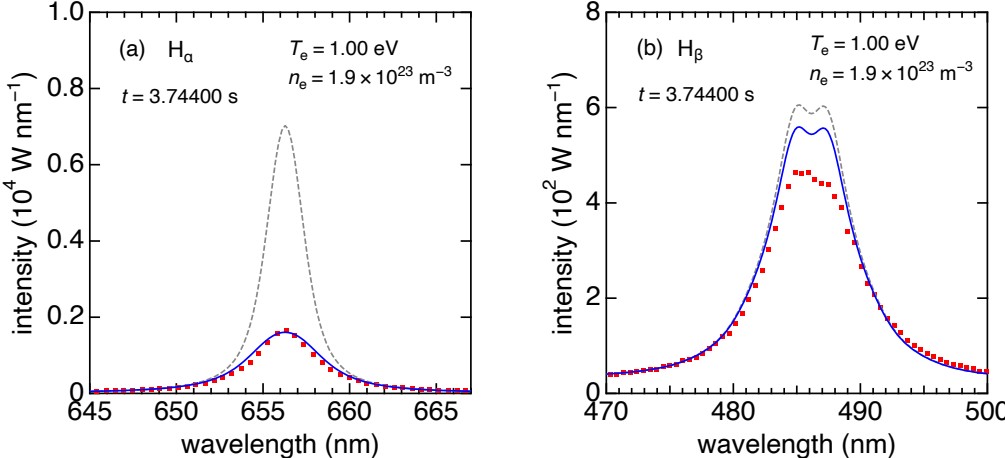

**Figure 9.** (**a**) Fitting result of the Balmer $\alpha$ line (solid line) with a model which takes into account the reabsorption effect for the measurement result in Figure 2c (red dots). The fitting result for the entire spectrum without the reabsorption effect is shown with the dashed line. (**b**) The synthetic profile for the Balmer $\beta$ line under the same absorption medium condition derived from the fitting of the Balmer $\alpha$ line. The red dots and the dashed line are the same meaning as those in (**a**).

The influence of the reabsorption effect for the Balmer $\beta$ line profile has also been evaluated. The optical thickness at the line center is evaluated to be 0.143 with the same $x$ value as obtained for the Balmer $\alpha$ line, and the resulting line profile is shown in Figure 9b. Although the influence of the reabsorption effect for the Balmer $\beta$ line is visible, the measurement result looks to be influenced more by the reabsorption effect. One clear reason for this inconsistency is that the model of the radiation transport is too simplified in the present analysis, i.e., the ablation cloud shape has an anisotropic structure and the plasma parameters in it should be non-uniform. No further analysis, however, is made here for the reabsorption effect of the Balmer $\beta$ line as it is not the main subject of this paper, but it should be at least noted that the Balmer $\beta$ line profile can be also significantly influenced by the reabsorption effect.

Lastly, it is worth noting that there is an ongoing discussion regarding the validity of OPF in high-density conditions, especially when perturbers penetrating into the wavefunction extent of a target atom play significant roles [20]. The wavefunction extent, approximately $p^2$ times the Bohr radius $a_0$, for $p = 6$, is roughly $1.9 \times 10^{-9}$ m, and the field strength at this distance is $4.0 \times 10^8$ V/m. When considering the Holtsmark distribution, the impact of such a strong field region on $w_\mathrm{f}$ is negligibly small, even with the highest $n_\mathrm{e}$, i.e., $2 \times 10^{23}$ m$^{-3}$, in the present study (see Figure 4). The effect of the wavefunction extent appears to be significant for cases with higher density conditions.

## 5. Summary

We have enhanced the spectral model by incorporating OPF, resulting in improved fitting results. In the practical analysis of the measured spectra, we have observed that the electron density in the ablation cloud is primarily determined by the background electron temperature. Additionally, there appears to be an upper limit for the electron density in the ablation cloud. These findings are expected to enhance the accuracy and reliability of the simulation model for pellet ablation, leading to a more comprehensive understanding of the ablation process.

On the other hand, as noted in Section 4, the theoretical model of the line profile used in the present analysis could be inappropriate especially in the density range higher than $10^{23}\,\mathrm{m}^{-3}$. We will consider employing FFM for improving the spectral model in future studies.

**Author Contributions:** Conceptualization, M.G., G.M., R.S., A.M. and B.P.; methodology, M.G., G.M. and B.P.; software, M.G. and B.P.; validation, M.G., R.S., G.M., A.M. and B.P.; formal analysis, M.G.; investigation, M.G.; resources, M.G.; data curation, M.G., T.O., T.K. and Y.K.; writing, M.G.; supervision, M.G. All authors have read and agreed to the published version of the manuscript.

**Funding:** This research was partly supported by JSPS KAKENHI Grant Number 18K03588 and by the National Institute for Fusion Science grant administrative budgets (ULHH028).

**Data Availability Statement:** The LHD data can be accessed from the LHD data repository at https://www-lhd.nifs.ac.jp/pub/Repository_en.html (accessed on 22 February 2022).

**Acknowledgments:** The authors thank all the LHD experiment group members for their support to conduct the experiment. M.G. thanks J.J. Simons for correcting the English.

**Conflicts of Interest:** The authors declare no conflicts of interest.

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
