# Peer review of "Better Understanding of Hydrogen Pellet Ablation Cloud Spectra through the Occupation Probability Formalism in LHD"

_atoms, doi:10.3390/atoms12010001_

Round 1
Reviewer 1 Report
Comments and Suggestions for Authors
The manuscript was performed at a high scientific level. The authors should consider improving the presentation of the material in Figure 4a (perhaps they should use a logarithmic scale also for y-axis) and swap the top and bottom axes in Figure 6.
Author Response
Thank you for your comments. We have revised the figures following your suggestions.
Reviewer 2 Report
Comments and Suggestions for Authors
The paper is devoted to experimental investigations of radiation spectra from pellet clouds in LHD device. The specific data are presented for Balmer series including their transition to the limit of Balmer series connected with continuum radiation. The modeling of such spectra are based on previuose method using an approximate presentation of emission spectra as a combination of static Hotzmark spectra multiplied on the occupation probability of energy levels determined by so called critical values of electric field strengths responsible for the cut of their contributions due to ionization in the electric field.
Of course, the experimental data from very prestige LHD installation are of permanent interest for readers. Especially it is the case for pellet cloud radiation emission being an exotic object of low temperature and high density plasmas inside the high temperature and low density surround plasma. In the context the important conclusion of the authors about strong correlations between pellet density and the temperature of surrounded plasma is of great interest for many investigators.
At the same time the theoretical modeling of hydrogen Balmer series on the basis of rather old theoretical models seems to be insufficient. The modeling deals with average estimations of critical electric fields which doesn’t depend on parabolic quantum numbers of excited hydrogenic atomic states in electric field. The more detail calculations of such effects are presented in the paper A. Calisti, L.A. Bureyeva, V.S. Lisitsa, D. Shuvaev, and B. Talin. Eur.Phys.J.D.v.42.387 (2007).
It seems to be natural to make a comparison with corresponding results.
In any case the present paper is of great interest for reader interested in atomic physics under such exotic conditions as pellet cloud radiation in thermonuclear plasmas.
So I can recommend the paper for publication in Atom after taking into account the comments pointed above.
Author Response
Thank you for your comment on the theoretical modeling used in the present manuscript. We have noticed that the Balmer γ line in Fig. 2 (f) is slightly broader than the synthetic spectrum while no such tendency is seen for the Balmer α and β lines. This inconsistency could be solved by adopting FFM. We have added the following paragraph in Sec. 4. We think to focus our interest on this subject in future studies.
"It should be noted here that this paper focuses on the method to smoothly connect discrete lines to the continuum in the Balmer series spectrum, and the data for the Stark broadening profiles remain the same in the present analysis. However, around the same time as Ref. [4], a new calculation method for the Stark broadening of the Balmer series lines has been developed [16,17]. This new method, called the frequency fluctuation method (FFM), has improved the treatment for the electron collision broadening by taking into account the gradual change from the impact approximation to the quasi-static approximation with increasing n_e, which typically leads to narrowing the line profile. Following the criterion for this transition discussed in Refs. [16,18], it is found that this effect for the Balmer α and β lines is found to be insignificant with the present n_e and T_e ranges, while the influence on the Balmer γ could be significant. Figure 2 (f) is the spectrum when n_e is relatively high, and the measured Balmer γ profile is found to be narrower than the synthetic spectrum. This difference could be explained by FFM, but further investigation would be necessary."
Reviewer 3 Report
Comments and Suggestions for Authors
I suggested a minor editing of English language required

Comments on the Quality of English LanguageAuthor Response
Thank you for your comments. The following is the list of our point-by-point replies to the comments.
1- The writing language of this manuscript is good, but some notes need to be revised.
The revised manuscript has been checked by a native English speaker, and we hope the English has been improved.
2- Please clarify the novelty and the extent of benefit of the article in the abstract and the introduction.
We have added the following sentences in the abstract and in the introduction, respectively.
"This type of correlation is first confirmed in the present analysis and should give a new insight in the simulation studies of pellet ablation for the magnetically confined fusion plasma."
"This kind of detailed spectroscopic analysis of the pellet ablation cloud is only performed at LHD, making this research highly unique. Improvement of the simulation model naturally leads to accuracy enhancement of derived plasma parameters in the pellet ablation cloud, which can be used for examining reliability of the pellet ablation simulation model. The results obtained in this study should therefore contribute to making clear the pellet ablation mechanism and hence enhancement of the particle fueling efficiency for the magnetically confined fusion plasma."
3- Please add the model, country of origin, specification, and operation conditions for the instruments used in the experiment setup, such as: CCD, spectrometer, lenses, and optical fibers.
We have added specifications of the optical fiber, spectrometer, and CCD in the manuscript. We do not use a lens for this measurement.
4- I suggested add the conclusion or summary section as individual section.
We have added a separate summary section as follows.
"We have enhanced the spectral model by incorporating OPF, resulting in improved fitting results. In the practical analysis of the measured spectra, we have observed that the electron density in the ablation cloud is primarily determined by the background electron temperature. Additionally, there appears to be an upper limit for the electron density in the ablation cloud. These findings are expected to enhance the accuracy and reliability of the simulation model for pellet ablation, leading to a more comprehensive understanding of the ablation process.
On the other hand, as noted in Sec. 4, the theoretical model of the line profile used in the present analysis could be inappropriate especially in the density range beyond 10^23 m^-3. We will consider employing FFM for improving the spectral model in future studies."
5- It was suggested that the manuscript be supported with new articles published between 2020 and 2023 to highlight the article's modernity.
We have added some new reference papers.
Reviewer 4 Report
Comments and Suggestions for Authors
Comment:
- Pallet or pellet, please spell check before submission.
- Is there any way to reduce or minimize re-absorption? Will the change of Balmer series to Lyman or Paschen series reduce the re-absorption?
- Do the authors think skipping some of the data points near the dip is the best way to get good fitting results?
- Initially, I still had 5 more comments or questions, but after I read p.290, especially Fig. 9, everything is explained now. All the “mysteries” are solved.
As mentioned in the last comment, the authors’ discussion is clear but too lengthy, other than that I think I don’t have any further comment.
Comments on the Quality of English Language
please spell check before submission.
Author Response
Thank you for your comments. We performed a spell check on the revised manuscript. Although the spell check was made also for the manuscript previously submitted, we have overlooked that you pointed out. A native English speaker has also checked the present manuscript.
As for the re-absorption effect, the Lyman series lines should be more influenced because the lower state of the Lyman series lines is the ground state which may have a much larger density than the lower state of the Balmer series lines, i.e., n = 2. The effect may be smaller for the Paschen series lines, but the measurement is difficult because they are in the near infrared wavelength range. I think the Balmer series lines are the best solution for this kind of measurement. We have skipped some data points in the central region of the Balmer alpha and beta lines in the fitting as a reasonable method, but it would be also possible to include optical thickness as a fitting parameter in the spectral model. We think to perform such an analysis in future studies. We have added a sentence in the first paragraph of Sec. 4 to let the readers know that this subject will be revisited later on in this paper.